# Safety and Effectiveness of Vedolizumab and Ustekinumab in Elderly Patients with Inflammatory Bowel Disease: A Real-Life Multicentric Cohort Study

**DOI:** 10.3390/jcm13020365

**Published:** 2024-01-09

**Authors:** Tom Holvoet, Marie Truyens, Cara De Galan, Harald Peeters, Francisco Mesonero Gismero, Ainara Elorza, Paola Torres, Liv Vandermeulen, Aranzazu Jauregui-Amezaga, Rocio Ferreiro-Iglesias, Yamile Zabana, Laia Peries Reverter, Jeroen Geldof, Triana Lobatón

**Affiliations:** 1Department of Internal Medicine and Pediatrics, Ghent University, 9000 Ghent, Belgium; marie.truyens@uzgent.be (M.T.); cara.de.galan@telenet.be (C.D.G.); jeroen.geldof@uzgent.be (J.G.); triana.lobatonortega@uzgent.be (T.L.); 2Ghent Gut Inflammation Group (GGIG), Ghent University, 9000 Ghent, Belgium; 3Department of Gastroenterology, VITAZ, 9100 Sint-Niklaas, Belgium; 4Department of Gastroenterology, UZ Ghent, Corneel Heymanslaan 10, 9000 Ghent, Belgium; 5VIB Center for Inflammation Research (IRC), Ghent University, 9000 Ghent, Belgium; 6Department of Gastroenterology, University Hospital Ghent, 9000 Ghent, Belgium; 7Department of Gastroenterology, AZ Sint Lucas, 9000 Ghent, Belgium; harald.peeters@azstlucas.be; 8Department of Gastroenterology, Hospital Ramon y Cajal, 28034 Madrid, Spain; pacomeso@hotmail.com; 9Department of Internal Medicine, Universidad de Alcalá de Henares, 28805 Madrid, Spain; 10Department of Gastroenterology, Hospital de Galdakao, 48960 Bilbao, Spain; ainara.elorzaechaniz@osakidetza.eus; 11Department of Gastroenterology, Hospital Universitari Germans Trias i Pujol, 08916 Barcelona, Spain; ptrodriguez@bsa.cat; 12Department of Gastroenterology, Vrije Universiteit Brussel (VUB), Universitair Ziekenhuis Brussel, 1090 Brussels, Belgium; 13Department of Gastroenterology and Hepatology, University Hospital Antwerp, 2650 Antwerp, Belgium; aranzazu.jaureguiamezaga@uza.be; 14Laboratory of Experimental Medicine and Pediatrics (LEMP), Division of Gastroenterology and Hepatology, Faculty of Medicine and Health Sciences, University of Antwerp, 2000 Antwerpen, Belgium; 15Department of Gastroenterology, Hospital Clínico Universitario de Santiago, 15706 Santiago de Compostela, Spain; rocioferstg@hotmail.com; 16Department of Gastroenterology, Hospital Universitari Mútua de Terrassa, 08221 Terrassa, Spain; yzabana@gmail.com; 17Centro de Investigación Biomédica en Red de Enfermedades Hepáticas y Digestivas (CIBERehd), 28029 Madrid, Spain; 18Department of Gastroenterology, Hospital Universitari de Girona, 17007 Girona, Spain; lperies.girona.ics@gencat.cat

**Keywords:** elderly, IBD, vedolizumab, ustekinumab, safety

## Abstract

Background: Data on ustekinumab and vedolizumab in the elderly inflammatory bowel disease (IBD) population are limited. The aim of the current study was to assess the safety and effectiveness of both in an elderly real-life population. Methods: A multicentric retrospective study was performed on IBD patients who started vedolizumab or ustekinumab between 2010 and 2020. Clinical and endoscopic remission rates and (serious) adverse events (AE) were assessed. Results: A total of 911 IBD patients were included, with 171 (19%) aged above 60 (111 VDZ, 60 UST). Elderly patients treated with vedolizumab or ustekinumab had an increased risk for non-IBD hospitalization (10.5% vs. 5.7%, *p* = 0.021) and malignancy (2.3% vs. 0.5%, *p* = 0.045) compared to the younger population. Corticosteroid-free clinical (50% vs. 44%; *p* = 0.201) and endoscopic remission rates (47.9% vs. 31%, *p* = 0.07) at 1 year were similar. Comparing vedolizumab to ustekinumab in the elderly population, corticosteroid-free (47.9% vs. 31%, *p* = 0.061) and endoscopic remission rates (66.7% vs. 64.4%, *p* = 0.981) were similar. Vedolizumab- and ustekinumab-treated patients had comparable infection rates (13.5% vs. 10.0%, *p* = 0.504), IBD flare-ups (4.5% vs. 5%, *p* = 1.000), the occurrence of new EIMs (13.5% vs. 10%, *p* = 0.504), a risk of intestinal surgery (5.4% vs. 6.7%, *p* = 0.742), malignancy (1.8% vs. 3.3%, *p* = 0.613), hospitalization (9.9% vs. 11.7%, *p* = 0.721), and mortality (0.9% vs. 1.7%, *p* = 1.000). AE risk was associated only with corticosteroid use. Conclusions: Ustekinumab and vedolizumab show comparable effectiveness and safety in the elderly IBD population. Elderly IBD patients have an increased risk for non-IBD hospitalizations and malignancy compared to the younger IBD population, with corticosteroid use as the main risk factor.

## 1. Introduction

Inflammatory bowel diseases (IBD) are a group of chronic, auto-immune inflammatory disorders comprising Crohn’s disease (CD) and ulcerative colitis (UC), both of which are associated with increased morbidity and mortality and impaired health-related quality of life [1]. Although the exact aetiology of IBD remains elusive, the current paradigm is based on the occurrence of a dysregulated immune response to the gut microbiome in patients who are genetically susceptible [1]. The treatment of IBD, therefore, is mainly focused on suppressing this immune response, and although therapeutic options have tremendously expanded over the last decades, even the more recently commercialized ones are associated with a significant risk for opportunistic infections and other immune-related complications [1,2].

One population that is particularly at risk for these immunosuppression-related complications is the elderly. Up to twenty percent of patients with IBD are diagnosed after the age of 60, and taken together with the overall ageing of the population, this group of elderly IBD patients will continuously expand in the future [3,4,5]. Even though several studies have shown that IBD at a later age does not have a more benign course and is associated with an increased need for hospitalization and surgery, immunosuppressant medication is used less frequently, mainly due to concerns of infectious complications and cancers in this group of patients with increased comorbidities, polypharmacy, and frailty [5]. However, robust data about the use of immunosuppressants in the elderly population are sparse as this group is often underrepresented and often excluded from clinical trials [6,7].

Initial observational studies, however, show that the risk of opportunistic infections is indeed higher in elderly patients compared to their younger counterparts when treated with monoclonal antibodies against tumor necrosis factor-alpha (anti-TNF), including infliximab, adalimumab, golimumab, and certolizumab [8,9,10]. Data on efficacy and safety in the elder population of the newer biological therapies such as the gut-selective vedolizumab (VDZ, an alpha4 beta7 integrin antagonist) and especially the anti-IL12/23 antagonist ustekinumab (UST) are, however, still limited [11,12,13]. Both biologicals could provide a safer alternative to anti-TNF in the elderly population. The aim of the current study was to assess the safety and effectiveness of the more recently commercialized agents VDZ and UST in a real-life elderly IBD population.

## 2. Methods

### 2.1. Patient Inclusion and Study Design

An international multicentric retrospective study was performed in 10 collaborative centres in Belgium and Spain. Patients with IBD who initiated VDZ or UST treatment between May 2010 and December 2020 at one of the participating centres were identified through hospitals’ medical records. All patients were at least 18 years of age at treatment initiation and were previously diagnosed with CD or UC (IBD-undifferentiated cases were excluded). Elderly IBD was defined as IBD in patients with a minimal age of 60 years [6]. A minimal follow-up duration of 8–14 weeks was required unless treatment was stopped earlier due to the development of complications or clear non-responsiveness to treatment. No other exclusion criteria were implemented.

### 2.2. Data Collection

Patient demographic and clinical characteristics were retrospectively collected from the electronic patients’ records and included age, gender, age at diagnosis, height, weight, baseline comorbidities, disease location, surgical treatment for IBD, previous use of biologicals, concomitant drug use, smoking behaviour, adverse events, and reason to stop treatment. Retrospectively, data on adverse events were collected starting from the date of first appearance.

### 2.3. Endpoints

Primary endpoint of the study was corticosteroid-free remission at 1 year. Secondary endpoints were corticosteroid-free remission at month 6, endoscopic remission at 1 year and overall safety.

#### 2.3.1. Assessment of Effectiveness

Clinical and endoscopic remission data were retrospectively collected at time points of 6 months, 1 year, and 2 years and were based on the treating physician’s assessment. Corticosteroid-free remission was defined as the presence of clinical remission at the time of evaluation without the use of corticosteroids.

#### 2.3.2. Assessment of Safety

Comorbidities present at treatment start were noted by the treating physician, and retrospectively, the Charlson comorbidity index (CCI) was calculated [14]. Attending gastroenterologists collected all data and potential adverse or serious adverse events (AEs and SAEs) that occurred during treatment with UST and/or VDZ from the date of first appearance. Adverse events were defined as the occurrence of extra-intestinal manifestations (EIMs), infections (including COVID-19), skin disorders, and gastrointestinal complications (e.g., (sub)obstruction). Serious adverse events were defined as intestinal resection, hospitalization (for IBD-flare or for an indication other than IBD), diagnosis of malignancy during treatment, or mortality [9].

### 2.4. Statistical Analyses

All analyses were performed in SPSS Statistics version 28 (IBM Corp. Released 2020. IBM SPSS Statistics for Windows, Version 27.0, Armonk, NY, USA). GraphPad Prism^®^ (GraphPad Software Inc., San Diego, CA, USA) was used for graphical representations of data. Descriptive statistics were presented as means ± standard deviations (SD) for continuous variables with a normal distribution, medians with interquartile ranges (IQR) for data with a non-normal distribution, and percentages for categorical variables. Chi-square or Fisher’s exact test was performed to compare categorical variables between the different treatments; continuous variables were analysed with the independent sample *t*-test (or Mann–Whitney U test in case of non-normal distribution). A two-tailed *p* < 0.05 was considered statistically significant.

Next, Cox proportional hazard models were used to assess the appearance of (S)AEs and the time of occurrence. Patients without (S)AEs were censored to the date of treatment discontinuation or date of last follow-up in case of treatment continuation. Covariate selection was based on clinical relevance and risk factors reported in the literature. Since this was a retrospective study and treatment selection (UST vs. VDZ) was not performed randomly, propensity score-weighted analyses were performed. The propensity score was calculated using a multivariate logistic regression model, including factors that could have influenced the selection of either VDZ or UST and the outcomes. This included age, IBD type, previous TNF-inhibitor use, and the country of data assessment. All results of the survival analyses are reported as hazard ratios (HR) and 95% confidence intervals (95% CI).

### 2.5. Ethical Statement

The study protocol was reviewed and approved by the Ethical Committee of the University Hospital of Ghent (EC2019-0978) as coordinating centre and obtained approval per requirements for each of the participating centres, following the Declaration of Helsinki.

### 2.6. Data Availability Statement

The data that support the findings of this study are available from the corresponding author upon reasonable request.

## 3. Results

### 3.1. Characteristics of the Study Population

In total, 911 patients with IBD were included in this multicentric retrospective cohort study, of whom 171 (19%) were aged 60 or older and considered elderly; in 33% of them, IBD was diagnosed after the age of 60 (elderly onset). In the entire cohort, 584 patients were treated with VDZ and 327 with UST. Baseline characteristics between the elderly and younger population were comparable, although there was a female predominance and less previous anti-TNF exposure among the elderly (Table 1).

In the elderly population, 111 patients were treated with VDZ and 60 with UST. Baseline characteristics such as age, sex, and disease location were similar between VDZ- and UST-treated patients (Table 2). In patients with CD, an equal distribution among both treatments was seen, whereas in UC, most patients were treated with VDZ and only a minority with UST (86.7% vs. 13.3%, *p* < 0.001), a difference presumably due to UST only being approved for the treatment of UC in 2020 (Belgium) or 2021 (Spain). In general, the UST cohort consisted of patients with more refractory disease, as reflected by the percentage of previous anti-TNFα use and other biologicals (33%), which was significantly higher compared to that in the VDZ-treated group (*p* < 0.001). Patients treated with VDZ received steroids more frequently at treatment start compared to patients treated with UST (58.6% vs. 28%, *p* < 0.001). Vedolizumab was used more often as a first- or second-line treatment (75%) as opposed to ustekinumab (55%), which was used more often as a third-line option.

Comorbidities, as measured by the Charlson comorbidity index, were similar between the UST- and VDZ-treated groups. As to be expected in an elderly population, the majority of patients had moderate to severe comorbidity (defined as a Charlson index ≥ 2), respectively, 55% and 47% in the VDZ and UST groups.

### 3.2. Safety of Vedolizumab and Ustekinumab Treatment in the Elderly Population

When comparing the safety profile of UST or VDZ treatment in the elderly population, no significant increase in infections (12.3% vs. 8.8%, *p* = 0.159) in general was observed. Specifically, the number of COVID-19 flares (0.5% vs. 3.2%, *p* = 0.273), urinary tract infections (0.9% vs. 0.9%, *p* = 0.412), C difficile infections (1.2% vs. 1.7%, *p* = 0.164) were comparable between groups; however, more non-COVID-19 respiratory infections were observed in the elderly population (6.6% vs. 2.5%, *p* = 0.01). No difference in IBD flare-ups (4.7% vs. 5.7%, *p* = 0.606) or new onset of extra-intestinal manifestations (EIMs) (12.3% vs. 11.6%, *p* = 0.814) was observed compared to the younger population. However, elderly patients were at an increased risk for non-IBD-related hospitalizations (10.5% vs. 5.7%, *p* = 0.021) when treated with UST or VDZ, compared to the younger population, even when correcting for previous anti-TNF use and disease type in a Cox regression model (Figure 1). Similarly, the adjusted risk of developing malignancy was significantly increased in the elderly population (2.3% vs. 0.5%, *p* = 0.045). Discontinuation rates of UST or VDZ were similar in both the younger and elderly population (31% vs. 25%) (*p* = 0.110). In the Cox regression model, age was not a contributing factor for drug discontinuation (HR 0.91 (0.66–1.23) *p* = 0.556). Previous anti-TNF use, however, was (HR 2.16 (1.28–3.63) *p* = 0.004).

When comparing the head-to-head safety of VDZ and UST in the elderly population, no significant differences were found between either treatments. Infection rates (13.5% vs. 10.0%, *p* = 0.504) in general were similar between VDZ- and UST-treated elderly patients, while specifically there was no difference in COVID-19 (0.9% vs. 0.0% *p* = 0.548), gastro-intestinal (3.6% vs. 3.3%, *p* = 0.990), urinary tract (0.9% vs. 0.0%, *p* = 0.645), and non-COVID-19 respiratory infections (7.2% vs. 6.8% *p* = 0.999). IBD flare-ups (4.5% vs. 5%, *p* = 1.000) and the occurrence of new EIMs (13.5% vs. 10%, *p* = 0.504) were all similar between VDZ- and UST-treated elderly patients. Similarly, there was no difference in the risk of intestinal surgery (5.4% vs. 6.7%, *p* = 0.742), malignancy (1.8% vs. 3.3%; *p* = 0.613), hospitalization (9.9% vs. 11.7%, *p* = 0.721), and mortality (0.9% vs. 1.7%, *p* = 1.000) between both groups. In a Cox regression model, the overall risk for adverse events was associated only with corticosteroid use but not with age or comorbidity score (Figure 2). Numerically, elderly patients treated with VDZ had a higher risk of stopping treatment compared to UST-treated patients, although the difference was not statistically significant (29.7% vs. 16.7%, *p* = 0.06).

### 3.3. Effectiveness of Vedolizumab and Ustekinumab in the Elderly Population

A corticosteroid-free clinical remission at 1 year was achieved in 44.5% (281/632) of patients in the younger cohort compared to 50.4% (71/141) in the elderly patients (*p* = 0.204) (Figure 3). Dose optimisation occurred in 14.4% of patients treated with VDZ and in 20.9% of UST-treated patients, while respectively, 11.6% and 14.6% of patients were on combined treatment with an immunosuppressant. In a multivariate logistic regression model, age was not significantly associated with achieving corticosteroid-free clinical remission (OR 0.23 (0.00–192.36), *p* = 0.66). The prior use of anti-TNF, however, as well as treatment with UST, were significantly associated with a lower chance of achieving corticosteroid-free clinical remission in both populations. Endoscopic remission at 1 year was achieved in 31% of younger patients compared to 47.9% in the elderly cohort (*p* = 0.07).

In the elderly cohort specifically, there was no statistically significant difference in steroid-free clinical remission at 1 year between the VDZ- and UST-treated patients (respectively, 55.8% (53/95) and 39.1% (18/46) (*p* = 0.064), although a significantly higher proportion of VDZ patients was in steroid-free clinical remission at month 6 (52/101 (51.5%) vs. 16/53 (30.2%), respectively)). In a sub-analysis comparing elderly UC and CD patients, no statistically significant differences in corticosteroid-free remission or endoscopic remission were found (Appendix A). Dose optimisation occurred in 7.9% of VDZ-treated patients as opposed to 23.7% of patients treated with UST, and combination therapy with an immunosuppressant occurred in, respectively, 9.8% and 7.5%. In a multivariate logistic regression model, no risk factors were significantly associated with attaining steroid-free remission at both time points (Appendix A). At 1 year, endoscopic remission rates were similar between both groups, respectively, 66.7% (60/90) and 64.4% (29/45) (*p* = 0.981) for the VDZ- and UST-treated patients (Figure 4).

## 4. Discussion

Elderly patients are an important and constantly growing subpopulation of IBD that is particularly vulnerable to both disease-related complications and side effects related to immunosuppressive treatment [3,4,6,15,16,17,18,19]. Conversely, it is a population that is systematically excluded from clinical trials, so data about which treatments are safe to use are scarce, especially regarding the newer biologicals. Real-life observational studies, such as the multicentric, multinational retrospective cohort study presented here, are of vital importance to understanding the risks and benefits of new treatments in these vulnerable populations.

In older patients, the course of IBD is not always more benign than in the young. On the contrary, the risk of disease-related complications such as surgery, hospitalization, and mortality can be even more substantial [15,20]. However, observational studies have shown that this group is often undertreated and subjected to long-term corticosteroid use due to concerns about the safety of steroid-sparing therapy in the elderly, leading to an even greater increase in morbidity and mortality [6,15,16].

Of all biologics, most data are available for anti-TNF and indeed confirm the theoretical concerns of increased infectious complications in these elderly patients: in an Italian multicentric cohort of 95 patients aged above 65 years, 11% developed severe infections, and 10% of these died because of them [21]. Other reports confirm an increased risk of infectious complications, and this is often a reason for stopping anti-TNF therapy in this patient group [22].

Vedolizumab, an alpha4beta7 integrin antagonist, theoretically holds advantages over other biologicals due to its localized gut action mechanism, lessening the impact of systemic immunosuppression. Indeed, post hoc analyses of the GEMINI registration trials did not show an increased risk of infections, albeit only a small proportion of patients included were older than 60 years [11]. Additional observational studies confirm the low risk of infectious and other therapy-related complications of vedolizumab in elderly patients; in two retrospective cohort studies, the risk of complications was deemed low, although the rate of complications with anti-TNF was comparable in these studies [10,23]. However, a retrospective study directly comparing anti-TNF and vedolizumab in elderly patients did find vedolizumab treated patients to have fewer infection-related hospitalizations compared to patients treated with anti-TNF [24]. A large US-based retrospective study found similar findings but only in patients with increased comorbidity (defined as the Charlson comorbidity index > 1) [25]. This study of 171 elderly IBD patients adds to the body of evidence that vedolizumab is safe to use in the elderly population.

Ustekinumab, a monoclonal antibody to the p40 subunit of IL12 and IL23, had already been used extensively and safely in psoriasis and psoriatic arthritis before becoming commonly used in both CD (UNITI trial in 2016) [26] and UC (UNIFI trials in 2019) [27]. The long-term follow-up IM-UNITI trials showed no difference in complications compared to the placebo; however, these included only a minority of elderly patients [28]. Large real-life effectiveness studies such as the ENEIDA registry confirmed these safety findings in clinical practice, but again, elderly patients were underrepresented [29]. Recently, a large retrospective study by the same group showed similar results concerning effectiveness and safety in a group of 212 elderly CD patients treated with Ustekinumab [27]. This study adds to this body of evidence and additionally shows similar effectiveness and safety of ustekinumab in elderly UC.

Importantly, this study showed that age does not affect the clinical efficacy of both vedolizumab and ustekinumab treatment, with similar rates of steroid-free clinical remission achieved in the elderly cohort. Only prior anti-TNF use was significantly associated with a lower risk of remission, signalling a more difficult-to-treat population. Other retrospective studies, such as the LIVE study, which included 198 matched elderly IBD patients, have shown similar effectiveness of vedolizumab in CD but not UC in elderly patients. However, in this cohort, no difference in the effectiveness of vedolizumab between elderly and younger UC patients could be found [30]. Of note, and importantly, vedolizumab functioned in this cohort more often as a second-line treatment, whereas ustekinumab was more frequently used as a third-line agent (25% vs. 45%).

Despite the favourable safety profiles of both VDZ and UST in elderly IBD patients, the overall risk of complications in this population is increased compared to their younger counterparts. As shown in our study, the risk of developing malignancy is elevated in the patient group older than 60 years of age, especially when previously treated with anti-TNF. Additionally, elderly patients have a higher overall risk of hospitalization, which seems to be largely related to corticosteroid use, underlining the need for steroid-saving strategies in this population.

Due to the retrospective design, this study has several limitations, such as a potential bias in the number of reported complications. However, since both VDZ and UST were assessed retrospectively, the design is unlikely to have caused a bias in favour of either treatment. Next, there was an uneven distribution between UC and CD in UST patients due to the different timing of reimbursement of UST for UC, which has an impact on the follow-up time. Lastly, the endpoints presented here were based on a physician’s assessment and not on standardized indices. However, the current study has several strengths; due to the multicentric international collaboration, a large population could be assessed during a long follow-up of 2 years. This is also one of the largest real-life cohorts of elderly patients being assessed for the safety of VDZ and UST to date.

In conclusion, this large, multi-centric, multi-national retrospective study shows both UST and VDZ to have comparable efficacy and a favourable safety profile in elderly IBD patients despite their different mechanisms of action. Despite this, the elderly IBD population remains at an increased risk for developing malignancies and hospitalization compared to younger IBD patients, the latter being primarily associated with corticosteroid use.

## Figures and Tables

**Figure 1 jcm-13-00365-f001:**
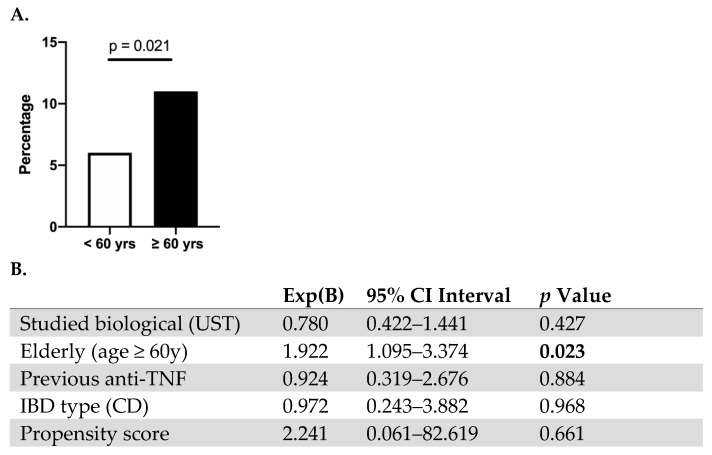
Elderly IBD patients treated with vedolizumab and ustekinumab are at increased risk for non-IBD-related hospitalization. (**A**) Rate of non-IBD-related hospitalization in the elderly (age ≥ 60 years) vs. younger population (<60 years) (**B**). Cox regression model with propensity score matching.

**Figure 2 jcm-13-00365-f002:**
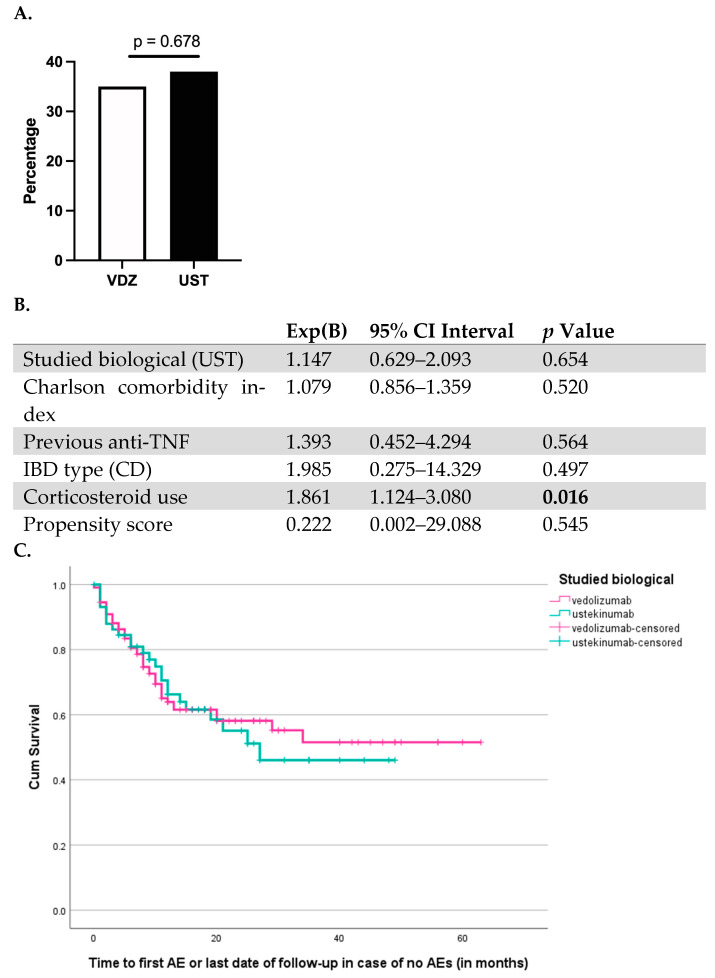
Risk of adverse events in the elderly population treated with vedolizumab and ustekinumab. (**A**) Percentage of elderly patients with adverse events in the VDZ- and UST-treated group (**B**) Cox regression model met propensity score patching (**C**). Cox regression survival curve.

**Figure 3 jcm-13-00365-f003:**
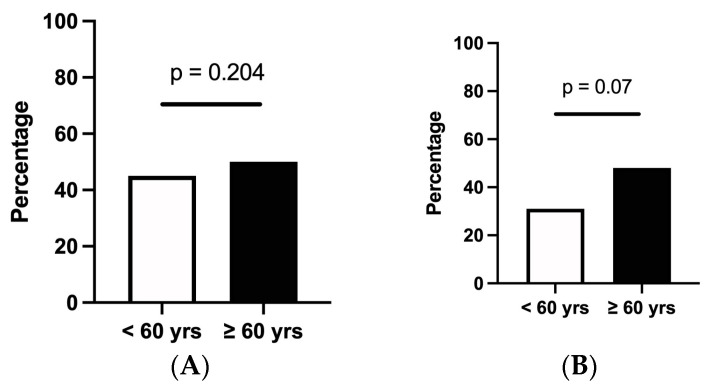
Effectiveness of vedolizumab and ustekinumab in the elderly population versus effectiveness in younger patients. (**A**) Corticosteroid-free clinical remission at 1 year after start of treatment with vedolizumab or ustekinumab (**B**). Endoscopic remission at 1 year after start of treatment.

**Figure 4 jcm-13-00365-f004:**
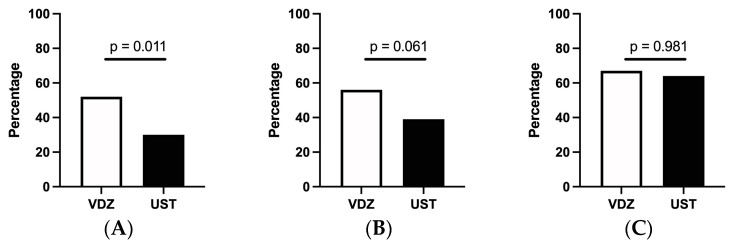
Effectiveness of vedolizumab and ustekinumab in the elderly population. (**A**) Corticosteroid-free clinical remission at 6 months (**B**). Corticosteroid-free clinical remission at 1 year (**C**). Endoscopic remission at 1 year. VDZ = vedolizumab, UST = ustekinumab.

**Table 1 jcm-13-00365-t001:** Baseline characteristics of the population.

	<60 Years Old (n = 740)	≥60 Years Old (n = 171)	Total (n = 911)	*p*-Value
**Age** (years)	39 (30–49)	66 (63–71)	43 (32–56)	<0.001
**Sex**				0.003
Female	400 (54.1)	71 (41.5)	471 (51.7)	
Male	340 (45.9)	100 (58.5)	440 (48.3)	
**Smoking behaviour ***				0.409
No	542 (78.3)	135 (82.3)	677 (79.1)	0.258
Current smoker	150 (21.7)	29 (17.7)	179 (20.9)	
**IBD type**				0.264
Ulcerative colitis	269 (36.4)	70 (40.9)	339 (37.2)	
Crohn’s disease	471 (63.6)	101 (59.1)	572 (62.8)	
**Disease duration** (years) *	9 (3–15)	11 (4–25.3)	9 (4–16)	<0.001
**Elderly onset IBD ***				<0.001
No	740 (100)	110 (66.3)	850 (93.8)	
Yes	0	56 (33.7)	56 (6.2)	
**BMI** (kg/m^2^) *	23.7 (21.1–26.9)	24.7 (21.9–27.9)	23.9 (21.1–27.1)	0.008
**Previous surgery**	235 (31.8)	48 (28.1)	283 (31.1)	0.348
**Montreal score for CD**				0.020
L1	126 (27)	42 (41.6)	168 (29.6)	
L2	114 (24.5)	25 (24.8)	139 (24.5)	
L3	212 (45.5)	32 (31.7)	244 (43.0)	
L3 + L4	14 (3.0)	2 (2.0)	16 (2.8)	
B1	201 (42.1)	41 (40.2)	242 (41.7)	
B2 B3	156 (32.6) 121 (25.3)	36 (35.2) 25 (24.5)	192 (33.1) 146 (25.2)	
**Montreal score for UC ***				0.937
E1	17 (6.3)	5 (7.2)	22 (6.5)	
E2	126 (46.8)	34 (49.3)	160 (47.3)	
E3	105 (39)	26 (37.7)	131 (38.8)	
Pouchitis	21 (7.8)	4 (5.8)	25 (7.4)	
**Previous use of biologicals**				
TNF antagonists	596 (80.5)	108 (63.2)	704 (77.3)	<0.001
Vedolizumab	94 (12.7)	20 (11.7)	114 (12.5)	0.720
Ustekinumab	20 (2.7)	1 (0.6)	21 (2.3)	0.152
Tofacitinib	13 (1.8)	1 (0.6)	14 (1.5)	0.488
**Comorbidities**				
COPD	0 (0.0%)	6 (3.5%)	6 (0.7%)	
Arterial hypertension	25 (3.4%)	9 (5.2%)	34 (3.7%)	
Cardiomyopathy	15 (2.0%)	4 (2.3%)	19 (2.1%)	
Diabetes	23 (3.1%)	2 (1.1%)	25 (2.7%)	
Obesity	10 (1.3%)	1 (0.5%)	11 (1.2%)	
Chronic kidney disease	0 (0.0%)	11 (6.4%)	11 (1.2%)	
Previous Cancer	30 (4.1%)	13 (7.6%)	43 (4.72%)	

Data are represented as median (IQR) or n (%). * Missing data: smoking n = 55, disease duration, and elderly onset IBD n = 5, BMI n = 181, Montreal score for UC n = 1.

**Table 2 jcm-13-00365-t002:** Baseline characteristics of the elderly population.

	Vedolizumab (n = 111)	Ustekinumab (n = 60)	Total (n = 171)	*p*-Value
**Age** (years)	67 (63–73)	65 (62–70)	66 (63–71)	0.159
**Sex**				0.977
Female	46 (41.4)	25 (41.7)	71 (41.5)	
Male	65 (58.6)	35 (58.3)	100 (58.5)	
**Smoking behaviour ***				0.409
No	90 (84.1)	45 (78.9)	135 (82.3)	
Current smoker	17 (15.9)	12 (21.1)	29 (17.7)	
**IBD type**				<0.001
Ulcerative colitis	62 (55.9)	8 (13.3)	70 (40.9)	
Crohn’s disease	49 (44.1)	52 (86.7)	101 (59.1)	
**Disease duration** (years) *	11 (4–25)	14 (7–26)	11 (4–25.3)	0.145
**Elderly onset IBD ***				
No	67 (62)	43 (74.1)	110 (66.3)	0.116
Yes	41 (38)	15 (25.9)	56 (33.7)	
**BMI** (kg/m^2^) *	9524.8 (22.3–28.3)	24.0 (21.9–27.0)	24.7 (21.9–27.9)	0.349
**Previous surgery**	28 (25.2)	20 (33.3)	48 (28.1)	0.260
**Montreal score for CD**				0.842
L1	20 (40.8)	22 (42.3)	42 (41.6)	
L2	14 (28.6)	11 (21.2)	25 (24.8)	
L3	14 (28.6)	18 (34.6)	32 (31.7)	
L3 + L4	1 (2)	1 (1.9)	2 (2)	
B1	20 (40.8)	21 (40.4)	41 (40.6)	
B2	18 (36.7)	18 (34.6)	36 (35.6)	
B3	11 (22.4)	13 (25)	24 (23.8)	
**Montreal score for UC ***				0.200
E1	5 (8.2)	0	5 (7.2)	
E2	28 (45.9)	6 (75)	34 (49.3)	
E3	25 (41)	1 (12.5)	26 (37.7)	
Pouchitis	3 (4.9)	1 (12.5)	4 (5.8)	
**Previous use of biologicals**				
TNF antagonists	62 (55.9)	46 (76.7)	108 (63.2)	0.007
Vedolizumab	-	20 (33.3)	20 (11.7)	-
Ustekinumab	1 (0.9)	-	1 (0.6)	-
Tofacitinib	0	1 (1.7)	1 (0.6)	0.351

Data are represented as median (IQR) or n (%). * Missing data: smoking n = 7, disease duration, and elderly onset IBD n = 5, BMI n = 35, Montreal score for UC n = 1.

## Data Availability

No new data were created or analyzed in this study. Data sharing is not applicable to this article.

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
