# Peer review of "Safety and Effectiveness of Vedolizumab and Ustekinumab in Elderly Patients with Inflammatory Bowel Disease: A Real-Life Multicentric Cohort Study"

_jcm, 2024, doi:10.3390/jcm13020365_

Round 1

Reviewer 1 Report

Comments and Suggestions for Authors

This is a retrospective observational study to compare safety and effectiveness of Vedolizumab and Ustekinumab in elderly patients with inflammatory bowel disease. The topic is very interesting, hovewer I have some concerns. The major are:

1.     The time interval of the study is 10 years, and the number of elderly patients is 171 out of the 10 participating centers, with 111 patients in vedolizumab (mainly 1st and 2nd line) and 60 in ustekinumab (3rd line) considering both patients with Crohn's disease and ulcerative colitis. It seems to me that the number of patients is not particularly large and that heterogeneity must be considered and indeed is a strong limitation of the study

2.     The study does not specify how clinical remission and endoscopic remission were assessed: e.g. MPS or HBI, SES-CD, MES... It would be important to know if the evaluation was done homogeneously over time and between the various participating centers

3.     The study does not include a sub-analysis of outcomes in Crohn's and ulcerative colitis separately, which in my opinion can be an interesting data to evaluate in a study that includes the two populations

Author Response

Thank you very much for your constructive comments. Please find the point-by-point answers to your questions attached.

Kind regards

Tom Holvoet

Reviewer 2 Report

Comments and Suggestions for Authors

The article " Safety and effectiveness of vedolizumab and ustekinumab in elderly patients with inflammatory bowel disease: a real-life multicentric cohort study" is a well-studied multi-centric study with significant findings. This paper is timely and informative, addressing a relevant topic in the literature. This study will serve as another addition in different patient cohort studying the effectiveness and efficacy of Vedolizumab (VDZ) and Ustekinumab (UST) therapy i.e. in the elderly Inflammatory Bowel Disease (IBD) population. The study is important as reference for the elderly IBD patient care management. This manuscript would be a stepping stone for the elderly IBD patient patient care at the time new effective therapies are being looked after.

The manuscript is well-written displaying results/information in targeted and well-designed fashion after thorough analysis. The available research is clearly presented, discussed, and the conclusion is supported by the evidence presented. The paper is very interesting.

A few minor suggestions for improving the manuscript are as follows:

1.    Please incorporate the full form of IBD in the first line of the abstract.

Author Response

Thank you very much for these constructive comments. Please find the point-to-point answers to your questions attached.

Kind regards

Reviewer 3 Report

Comments and Suggestions for Authors

The manuscript reports a retrospective study concerning the safety and efficacy of ustekinumab and vedolizumab in elderly inflammatory bowel disease patients, as compared to young ones.

Real world data are important in this context.

However, the Authors do not clearly define in the methods what are the primary and the secondary outcomes of the study.

The main limitation of this study is that both clinical and endoscopic remission are defined according to physician's assessmement. This must be acknowledged.

The Authors should cite other important recent papers in the field like Aliment Pharmacol Ther. 2022;56:95–109.

Author Response

Thank you very much for your review and comments.

Please find our point-to-point response attached.

Round 2

Reviewer 1 Report

Comments and Suggestions for Authors

The limitations of the study are still important despite the corrections. The subjectivity of patients assessment  for example is a limit that cannot be exceeded.